# Boosting Lipofection Efficiency Through Enhanced Membrane Fusion Mechanisms

**DOI:** 10.3390/ijms252413540

**Published:** 2024-12-18

**Authors:** Rais V. Pavlov, Sergey A. Akimov, Erdem B. Dashinimaev, Pavel V. Bashkirov

**Affiliations:** 1Research Institute for Systems Biology and Medicine, 18 Nauchniy Proezd, Moscow 117246, Russia; 2Frumkin Institute of Physical Chemistry and Electrochemistry, Russian Academy of Sciences, 31/4 Leninskiy Prospekt, Moscow 119071, Russia; akimov_sergey@mail.ru; 3Center for Precision Genome Editing and Genetic Technologies for Biomedicine, Pirogov Russian National Research Medical University, Moscow 117997, Russia; dashinimaev@gmail.com

**Keywords:** nanotechnology, transfection, lipid-based vector, membrane fusion, fusion pore, cationic lipids

## Abstract

Gene transfection is a fundamental technique in the fields of biological research and therapeutic innovation. Due to their biocompatibility and membrane-mimetic properties, lipid vectors serve as essential tools in transfection. The successful delivery of genetic material into the cytoplasm is contingent upon the fusion of the vector and cellular membranes, which enables hydrophilic polynucleic acids to traverse the hydrophobic barriers of two intervening membranes. This review examines the critical role of membrane fusion in lipofection efficiency, with a particular focus on the molecular mechanisms that govern lipoplex–membrane interactions. This analysis will examine the key challenges inherent to the fusion process, from achieving initial membrane proximity to facilitating final content release through membrane remodeling. In contrast to viral vectors, which utilize specialized fusion proteins, lipid vectors necessitate a strategic formulation and environmental optimization to enhance their fusogenicity. This review discusses recent advances in vector design and fusion-promoting strategies, emphasizing their potential to improve gene delivery yield. It highlights the importance of understanding lipoplex–membrane fusion mechanisms for developing next-generation delivery systems and emphasizes the need for continued fundamental research to advance lipid-mediated transfection technology.

## 1. Introduction

The precise delivery of genetic material into targeted eukaryotic cells, a process known as transfection, represents a pivotal approach with vast potential for elucidating genetic cellular processes and offering innovative solutions for regenerative medicine, hereditary diseases and cancer [1,2,3,4]. The introduction of foreign nucleic acids, including deoxyribonucleic acid (DNA), ribonucleic acid (RNA), messenger RNA (mRNA), small interfering RNA (siRNA), short hairpin RNA (shRNA), and microRNA (miRNA), into mammalian cells enables precise manipulation of gene expression, functional protein characterization, and the production of essential proteins for diverse applications [5,6,7,8,9] (Figure 1).

A major challenge in nucleic acid delivery is the development of vectors that are both safe and highly effective at overcoming biological barriers, both extracellular and intracellular [10]. Extracellular challenges encompass nuclease activity, interactions with serum proteins, endothelial barrier penetration, and target cell localization. Once these barriers are navigated, intracellular hurdles arise due to the polyanionic nature of nucleic acids, which prevents them from passing through the hydrophobic core of the lipid bilayer and reaching the cytoplasm.

External forces can be leveraged to breach cellular barriers and facilitate the delivery of nucleic acids into the cytoplasm. Techniques such as electroporation [11], microinjection [12], sonoporation [13], and a rapid decrease in hydrostatic pressure [14] have been shown transiently to disrupt cell membrane integrity, enabling the entry of genetic material [3,6,15]. These physical methods are reliable, fast, and efficient, making them indispensable in various areas of fundamental biological research [5,6]. However, they do compromise cell integrity, exposing the cytosol to potential pathogens present in the media [6,11], which can result in cell damage or even cell death [16]. Additionally, these methods face limitations, such as being impractical for in vivo applications and requiring specialized equipment and expertise to execute [17].

In the field of gene transfection, two primary types of vectors are commonly employed: viral and non-viral. Viral vectors have made significant strides in delivering genetic material for both transient and permanent transfection [18]. However, they present concerns including the risk of carcinogenesis [19], immunogenicity [20,21], broad or restricted tropism [22], limited DNA packaging capacity [23], and production challenges [24]. In contrast, non-viral vectors, particularly synthetic vectors, exhibit lower immunogenicity [25] and offer a promising alternative for addressing the safety concerns associated with viral vectors. The use of the CRISPR/Cas9 system enhances stable transfection and significantly increases the potential applications of non-viral vectors [26]. Among non-viral gene delivery methods, lipid-based vectors have emerged as versatile and effective tools for facilitating the transfer of genetic cargo due to their biocompatibility, ease of formulation, and ability to encapsulate a wide range of nucleic acids [27]. This year marks the 60th anniversary of Bangham’s pioneering work in 1964 [28,29], which first demonstrated the potential of lipid nanoparticles for drug delivery, laying the foundation for modern lipid-based gene therapy approaches. Nevertheless, their clinical development has been somewhat constrained by lower delivery efficiency compared to viral vectors [25].

Various types of lipid-based vectors, such as liposomes, solid lipid nanoparticles (SLNs), and nanostructured lipid carriers (NLCs), have been developed for gene transfection. These vectors have gained significant attention, especially in the 21st century [30]. The recent success of lipid-based COVID-19 vaccines from Pfizer/BioNTech and Moderna [31] has highlighted the potential of lipid-based transfection systems, underscoring their biocompatibility and membrane-mimicking abilities. Despite their structural diversity, all lipid-based vectors share a fundamental characteristic: the necessity for fusion between the cellular membrane and the lipid shell of the vector to achieve effective transfection. This fusion process is pivotal for the successful internalization and subsequent expression of exogenous genetic material within the target cell. Thus, regardless of their structural differences, the membrane fusion process is a critical determinant of the transfection efficiency of lipid-based vectors. Enhancing membrane fusion presents a promising avenue to improve the efficacy and specificity of lipid-based transfection systems.

The current understanding of the complex mechanisms and factors that regulate membrane fusion provides a foundation for optimizing these lipid-based vectors, advancing both research and therapy. This review examines the role of membrane fusion in determining the efficiency of lipid-based vectors and its potential to enhance gene transfection outcomes.

## 2. Lipid-Based Vectors for Gene Delivery: From Classical Liposomes to Emerging Technologies

The development of lipid-based vectors for gene transfection has been a transformative advancement in biotechnology and therapeutics. The initial proposal of liposomes as gene delivery vehicles in the late 1970s [32,33] was soon followed by the creation of the first cationic lipids, which form unilamellar liposomes capable of naturally complexing with negatively charged DNA and RNA molecules [34,35]. These innovations laid the groundwork for exploiting lipid properties to encapsulate genetic material and facilitate its cellular transfer. Over time, extensive research and technological advancements have refined lipid-based vectors, leading to novel formulations like solid lipid nanoparticles, nanostructured lipid carriers, cubosomes, and hexosomes, each providing distinct benefits for gene transfection applications (Figure 2a) [36].

Liposomes are spherical, vesicular structures made of one or more lipid bilayers surrounding an aqueous core, ideal for encapsulating nucleic acids. They can be classified into unilamellar and multilamellar structures, each with specific advantages for nucleic acid delivery [37,38]. Unilamellar liposomes have a single bilayer, while multilamellar liposomes feature multiple bilayers separated by aqueous spaces. Nucleic acids can be encapsulated within liposomes when DNA or RNA is dispersed in an aqueous medium during the formation process [39,40]. As lipids self-assemble into bilayers, nucleic acids are trapped within the aqueous compartments (Figure 2(ai))—a method suitable for smaller nucleic acids or oligonucleotides but less effective for larger molecules due to limited encapsulation efficiency.

The advent of cationic lipids has greatly facilitated the entrapment of nucleic acids in lipid nanoparticles. The combination of cationic and helper lipids with DNA or RNA results in the spontaneous formation of lipoplexes, stable lipid liquid crystalline nanoparticles that encapsulate nucleic acid molecules [41,42]. The specific lipid composition, nucleic acid length, and surrounding aqueous environment determine whether lipids in lipoplexes form a lamellar phase—with multilamellar lipid layers [43], or a non-lamellar phase, such as inverted hexagonal phases and hexasomes bounded by a lipid monolayer [44,45,46], or the inverted bicontinuous cubic phase, known as a cubosome [47,48]. Consequently, nucleic acid molecules are located within the aqueous space between lipid bilayers [43] or within inverted micelles and water channels of non-lamellar lipid nanoparticles (Figure 2(ai,aii)) [46,49]. In addition to complexing with DNA and RNA, cationic lipids facilitate the adhesion and fusion of lipoplexes with cytoplasmic or endosomal membranes (Figure 2b), which is necessary for gene delivery into the cytoplasm. To date, a substantial number of cationic lipids have been synthesized [50], including ionizable variants that acquire a net positive charge at the low pH conditions of late endosomes while remaining neutral at physiological blood pH, significantly minimizing immune response and clearance from the bloodstream [51]. These lipids have proven effective in enhancing transfection through endosomal escape (Figure 2b) [52] forming the basis for commercial vectors used in both in vitro and in vivo applications. However, it is important to note that excessive incorporation of cationic or pH-activated ionizable lipids, which increase cellular uptake and endosomal escape, can cause unwanted toxicity due to membrane destabilization [53]. Therefore, lipid component selection and composition must be carefully considered to minimize potential toxic effects [54,55].

In addition to cationic/ionizable species, lipid-based vectors often include auxiliary lipids or lipid-like molecules (Figure 2a), such as phosphatidylcholines and cholesterol, to enhance lipoplex stability. Phosphatidylethanolamines contribute to endosomal membrane destabilization, facilitating lipoplex escape. Polyethylene glycol (PEG)-functionalized lipids or other stabilizers are utilized to form a polymer outer corona, which prevents lipoplex aggregation and serum protein fouling, leading to enhanced circulation times in the bloodstream by reducing opsonization and clearance by the reticuloendothelial system [56,57]. The degree of PEGylation and PEG molecular weight can be adjusted to strike a balance between protection and targeting, enabling vectors to exploit the enhanced permeability and retention effect to effectively reach tumor sites [58].

For transfection purposes, both DNA and RNA molecules can be encapsulated in SLNs [59]. Enveloped in a lipid monolayer, these nanoparticles have a solid core made from high-melting lipids, remaining solid below 50 °C. Typically composed of cationic, neutral, and helper lipids, plus surfactants, SLNs compact DNA while solid-lipid components self-assemble to form the core (Figure 2a). NLCs represent a sophisticated version of SLNs, incorporating a liquid lipid phase for greater internal flexibility, enhancing capacity, stability, and versatility [60]. They feature a more complex internal structure due to the incorporation of a liquid lipid phase (Figure 2(aiii)). This inclusion allows for a more flexible arrangement of internal components, enhancing the loading capacity, stability, and versatility of the particles. Initially successful for hydrophobic cargo, methods to encapsulate hydrophilic materials are in active development [61]. SLNs and NLCs typically offer better stability than lipoplexes and liposomes, as their solid or solid–liquid cores reduce the risk of leakage. This stability benefits long-term storage and controlled release; they can also be crafted with diverse lipids and surfactants for improved dispersibility and bioavailability [62].

Selective accumulation in specific organs or cells can be achieved by incorporating targeting molecules into lipoplexes. Surface modification with ligands such as antibodies or peptides allows for binding to specific receptors on target cells [63,64,65,66]. Additionally, altering the lipid composition with permanent cationic lipids like dioleoyl-3-trimethylammonium propane (DOTAP) or anionic lipids like 1,2-dioleoyl-sn-glycero-3-phosphate (18PA), without changing core component ratios, converts the lipoplexes into selective organ targeting (SORT) nanoparticles. SORT nanoparticles accumulate in target organs based on the type and quantity of supplemental SORT molecules [67,68]. The SORT methodology’s potential for achieving tissue tropism could revolutionize drug delivery using lipid-based vectors [69].

The advancement of lipid-based vectors for gene transfection has led to more effective and specific strategies. Notably, nanoghosts, or cell-derived nanovesicles, have emerged as promising lipid-based transfection vectors (Figure 2(aiv)). Both nanoghosts and exosomes are derived from entire cell membranes, either naturally secreted by cells [70] or prepared via external stimuli such as mechanical extrusion or ultrasonication [71]. These vesicles maintain biological surface proteins and lipids in a nanoscale format [72], enabling them to imitate the targeting capabilities of their source cells. This allows them to efficiently home in on tumor sites or specific tissues targeted by their parent cells [73].

Structurally, nanoghosts retain the semi-fluidity of their lipid bilayer, akin to liposomal systems, but offer enhanced stability due to their complex lipid composition. This complexity promotes dynamic interactions with cellular membranes, boosting cellular uptake and gene delivery [74,75]. Genetic material can be incorporated during nanoghost formation through electroporation [76] or by pre-loading parent cells before vesicle production, ensuring secure genetic payload containment. Recent studies have shown nanoghosts to be effective in delivering small interfering RNA (siRNA), plasmid DNA, and CRISPR/Cas9 systems, underscoring their versatility in genetic therapies [77]. Cells can be selected for creating nanoghosts to meet specific therapeutic needs, such as using cancer cell-derived nanoghosts to deliver anti-cancer genetic material to the same cancer tissue [78,79].

Despite the success of lipids in effective mRNA delivery, especially notable in COVID-19 vaccines from Pfizer and Moderna [31,80], lipid-based vectors still lag behind viral vectors in transfection efficiency. This disparity is expected given that viral proteins have evolved over millions of years to target and deliver genetic material into the cytoplasm or cell nucleus with remarkable precision [23,81]. However, the benefits of lipid-based vectors warrant continued research to enhance their ability to cross biological barriers. With the advent of CRISPR/Cas9 technology, stable transfection is no longer exclusive to viral vectors [26]. The transfection efficiency of lipoplexes in delivering genetic material is closely tied to their physicochemical properties, which influence their ability to traverse cellular membranes and release nucleic acids into the cytoplasm (Figure 2b).

Cell entry of lipoplexes occurs predominantly through endocytic pathways (Figure 2(bii,biii)), including clathrin-mediated endocytosis, caveolae-mediated endocytosis, and micropinocytosis. The relative contribution of each pathway depends on multiple factors, including lipoplex composition, size, surface charge, and cell type [82,83]. Following internalization, lipoplexes enter the endosomal trafficking network, where they encounter progressively acidifying environments during endosome maturation: from early endosomes (pH~6.5) to late endosomes (pH~5.5), and ultimately to lysosomes (pH~4.7), which are rich in nucleases [84,85]. Consequently, nucleic acids must escape from endosomes before they transition into lysosomes to avoid enzymatic degradation (Figure 2(bii,biii)) [86]. This endosomal escape is both the most challenging and least understood step, often proving inefficient and representing a major bottleneck in the transfection process [84].

Two main hypotheses describe the mechanisms of endosomal escape and nucleic acid release. The first hypothesis suggests that release occurs through electrostatic interactions between the vector’s positively charged lipid envelope and the negatively charged endosomal membrane. This interaction triggers endosomal membrane destabilization, primarily through its fusion with the vector’s lipid envelope, thereby creating a direct pathway for material transfer from the vector’s interior to the cytoplasm (Figure 2(bii)) [87]. The second hypothesis, known as the “proton sponge effect”, proposes that ionizable lipids with buffering capacities trigger increased proton pump activity and chloride ion influx into the endosome. This influx results in elevated osmotic pressure and subsequent endosomal swelling, ultimately rupturing the membrane and releasing its contents into the cytosol (Figure 2(biii)) [88].

While the role of the “proton sponge effect” in endosomal escape for lipid vectors remains incompletely characterized, experimental evidence demonstrates a clear positive correlation between vector fusogenicity and transfection efficiency. Studies utilizing specific inhibitors have provided crucial mechanistic insights: the membrane fusion inhibitor Z-Phe-Phe-Phe-OH significantly reduces transfection efficiency, while the endocytosis inhibitor Chlorpromazine shows minimal impact [89], highlighting fusion as a critical determinant of successful gene delivery. Notably, highly fusogenic lipid vectors can circumvent the endosomal pathway entirely through direct plasma membrane fusion (Figure 2(bi)) [89,90,91,92]. Indeed, when endocytic uptake was experimentally inhibited in studies with commercial lipofection agents, the forced shift toward plasma membrane fusion actually enhanced overall transfection efficiency [91], likely due to both avoiding endosomal degradation and enabling immediate cytoplasmic access.

The fusion process promotes successful cargo delivery through two distinct but complementary mechanisms: direct cytoplasmic delivery via membrane merger, and the intermixing of vector and cellular membrane lipids, which facilitates the dissociation of DNA/RNA from their cationic lipid complexes [93]. Crucially, this fusion-mediated delivery preserves membrane integrity, ensuring safe and effective transfection. Understanding these fusion mechanisms at the molecular level is therefore essential for rational lipid vector design, with optimal formulations requiring careful optimization of both the fusion energy barrier and controlled lipoplex disassembly to maximize cytoplasmic delivery of genetic cargo.

## 3. Mechanisms and Energetics of Membrane Fusion: From Molecular Principles to Biological Regulation

### 3.1. General Principles and Energy Landscape of Membrane Fusion

Despite variations in rate, compartment size, and fusion mediators involved, lipid membrane fusion across biological processes follows a conserved two-step sequence. After membranes are brought into close proximity, a series of distinct structural rearrangements of lipid bilayers occurs, ultimately leading to a complete membrane merger [94,95,96,97]. While the classical model of membrane fusion was developed for bilayers, molecular dynamics simulations have revealed that even non-lamellar lipoplexes follow similar fusion intermediates [98], suggesting universal mechanistic principles underlying membrane fusion. For a comprehensive exploration of these principles and their thermodynamic aspects, we recommend consulting the following reviews [94,99,100,101,102,103,104]. Below, we briefly summarize the current understanding of the fusion mechanism and the energetic barriers, which are crucial for enhancing the fusogenicity of lipid-based vectors.

The fusion pathway proceeds through several coordinated stages (Figure 3a). Initially, membranes must overcome both electrostatic and hydration barriers to achieve close proximity (Figure 3b) [95,105]. As membranes approach each other, hydration repulsion increases dramatically at distances below 2 nm [105,106], making membrane juxtaposition energetically feasible only at localized points. These contact points typically form through membrane bulging [107]. The energy barrier for a close approach can be reduced by the formation of small hydrophobic defects in proximal monolayers, created when polar lipid headgroups shift laterally from the contact region. These defects become particularly effective when they align coaxially at membrane bulges where the separation distance is less than 1 nm [108,109].

Once water-free local contact is established, membrane reorganization proceeds through distinct intermediate structures. The initial formation of a stalk—a non-bilayer, hourglass-shaped intermediate—requires a local transition from lamellar to hexagonal phase in proximal monolayers [95,97]. Radial stalk expansion brings hydrophobic tails of distal monolayers into contact, forming a hemifusion diaphragm. The hemifusion diaphragm exhibits distinct structural properties compared to conventional lipid bilayers. Structural studies and molecular dynamics simulations reveal that the hemifusion diaphragm is characterized by reduced thickness and diminished molecular ordering [110]. The tails of lipids involved in the hemifusion diaphragm are packed in a more perpendicular fashion with respect to the hemifusion diaphragm normal. The resulting configuration creates a metastable structure with increased susceptibility to local disruptions through the formation of fusion pores, completing the membrane merger and aqueous context mixing (Figure 3a).

The formation of membrane fusion intermediates requires substantial energy input, primarily because lipid bilayers and monolayers inherently resist deformation into high-curvature configurations. Free energy calculations along the fusion pathway reveal the stalk as a semi-stable intermediate, with overall fusion kinetics governed by the energy barriers associated with both stalk formation and its transition to a fusion pore (Figure 3b) [95,97,99]. The energy required for stalk formation typically ranges from 25 to 50 k_B_T, depending on membrane mechanical properties and geometric constraints [95,111,112,113]. The subsequent transition from stalk to fusion pore demands even higher energy [114], with continuum models predicting approximately 40 k_B_T for membranes composed of cylindrical lipids with zero spontaneous curvature [112,115]. While these energetic barriers prevent spontaneous fusion in biological systems, nature has evolved specialized protein machinery to overcome them. Understanding these biological solutions provides valuable insights for vector design.

### 3.2. Biological Fusion Machinery

In biological systems, membrane fusion does not occur spontaneously; it requires specific fusion proteins that mediate the process. These proteins ensure that membranes come into close proximity, stabilizing their alignment at distances sufficient to trigger structural remodeling [81,102,116]. A substantial body of theoretical and experimental evidence indicates that overcoming hydration repulsion is the most energy-intensive step in fusion. Once close contact is established, thermal fluctuations of the lipid bilayers allow the remaining fusion reaction to proceed spontaneously, provided the energy required does not exceed a critical threshold of approximately 40 k_B_T [95,101].

However, the role of fusion proteins extends beyond merely bridging membranes. Their fusogenic activity catalyzes lipid rearrangements in response to specific signals, such as changes in pH, calcium ion concentrations, or the presence of proteases [116,117,118]. An essential aspect of lipid bilayer destabilization involves generating elastic stress through these proteins [103,104,107,119,120,121,122,123]. This bending stress on membrane bulges at fusion sites enhances stalk formation by increasing the free energy of the initial state, thereby reducing the energy required to overcome fusion barriers [103,116,123,124,125,126]. Additionally, proteins can induce lateral tension within the hemifusion diaphragm, facilitating the formation of fusion pores [127,128]. By embedding amphiphilic or hydrophobic motifs into the membrane, these proteins alter lipid packing, destabilizing the bilayer and lowering the energy threshold for fusion [129,130]. Thus, fusion proteins effectively guide lipid bilayer remodeling along the fusion pathway by either elevating the initial energy state or reducing the thresholds for generating intermediate structures. This enhanced efficiency is one reason why viral vectors outperform lipid vectors, which lack such evolutionarily refined protein machinery.

### 3.3. Role of Lipid Structure and Composition in Membrane Fusion

The energetics of fusion heavily depend on the lipid composition of merging membranes. Molecular dynamics simulations reveal that variations in lipid composition can modulate the stalk formation barrier by approximately 36 k_B_T [131]. Lipids with pronounced negative spontaneous curvature, particularly dioleoyl-phosphatidylethanolamine (DOPE) with its inverted conical shape, significantly enhance stalk formation and membrane fusion [132,133,134]. DOPE functions through two mechanisms: inducing hydrophobic defects in proximal monolayers [135] and reducing bending stress in the stalk membrane through curvature–composition coupling, which concentrates phosphatidylethanolamine in concave membrane regions [136].

Cholesterol influences fusion through multiple pathways. Its inverted-cone shape helps fill membrane voids and stabilize negative curvature along the stalk rim [137]. While molecular simulations indicate that cholesterol may not concentrate within the stalk itself, it significantly reduces hydration repulsion, thereby lowering fusion barriers [131,138]. However, cholesterol’s effects are complex; it can increase membrane ordering in some lipid environments, potentially inhibiting fusion [139]. Thus, cholesterol’s impact on fusion depends strongly on the surrounding lipid composition. It is also important to recognize that an excess of negative curvature lipids (either DOPE or cholesterol) can be detrimental, as high concentrations in distal monolayers may inhibit fusion pore opening and arrest the process at hemifusion [140].

Conversely, the presence of cone-shaped lipids with positive spontaneous curvature, such as lysophosphatidylcholine (lysoPC), exhibits a distinctly different impact on fusion processes. The presence of cone-shaped lipids, particularly in proximal monolayers, decreases the likelihood of stalk formation [134,141,142]. However, when present in distal monolayers, these lipids promote the poration of the hemifusion diaphragm [143,144]. Thus, optimizing lipid composition with negative curvature lipids in proximal monolayers and positive curvature lipids in distal ones is expected to yield a high propensity for fusion [144,145,146]. Additionally, the thickness of the hemifusion diaphragm influences fusion progression, with molecular dynamics simulations indicating that longer lipid tails may impede fusion pore opening [98].

The degree of unsaturation in phospholipid fatty acyl groups significantly influences fusion competence. Unsaturated bonds create kinks in hydrocarbon chains, preventing tight packing and enhancing membrane fluidity. This increased fluidity reduces intermolecular order and improves membrane permeability [147]. Polyunsaturated lipids, with their multiple double bonds, provide configurational flexibility that facilitates membrane curvature adaptation while reducing bending rigidity. Both cellular and model membrane studies demonstrate enhanced membrane remodeling upon incorporation of polyunsaturated lipids [148]. This unsaturation-mediated flexibility proves crucial for maintaining the fluidity, disorder, and elasticity required for efficient progression through fusion intermediates [140,149,150].

### 3.4. Elastic Stress in Membrane Fusion

Despite the complex requirements for lipid composition in different fusion stages, elastic strain consistently emerges as a critical factor in enhancing fusogenicity. This elastic stress can arise from either membrane leaflet deviations from their spontaneous curvature (determined by conical-to-cylindrical lipid ratios) or from externally generated mechanical tension. Membrane tension plays a biphasic role: initially facilitating membrane docking by reducing thermal undulations, while potentially hindering stalk formation by increasing associated energy barriers [151]. However, during later stages, tension promotes fusion pore opening and expansion [114,127]. Experimental evidence consistently shows that increased lateral tension in either or both membranes enhances fusion probability [152,153,154], underscoring the pivotal role of elastic stress in catalyzing membrane fusion.

With this comprehensive understanding of membrane fusion dynamics, we can now explore innovative strategies to enhance lipofection efficiency through optimized lipid vector–membrane interactions. The following section will examine these strategies in detail, focusing on their potential to improve genetic material delivery into cells through enhanced membrane fusion.

## 4. Strategies to Enhance Fusogenicity of Lipid-Based Vectors

### 4.1. Optimizing Vector–Membrane Interaction: From Electrostatic to Molecular Zippers

To achieve fusion, lipid vectors must closely approach cellular membranes, establishing direct contact between the hydrophobic cores of their proximal monolayers. This proximity is crucial, as lipid leaflets must be within 1 nm to enable hydrophobic interactions essential for membrane merging [108]. Cationic or ionizable lipids, commonly employed in lipoplex formulations [37,50,80,155], leverage electrostatic attraction to overcome hydration repulsion with negatively charged cell membranes [156] and to bring membranes together at a distance sufficient to create a hydrophobic connection between lipid compartments (Figure 3a and Figure 4a).

Significant advances have been made in the design and synthesis of cationic lipids for gene delivery, supported by extensive analytical techniques that have revealed structure-activity relationships [87,157,158]. The chemistry of the headgroup serves multiple functions: mediating DNA complexation, facilitating cellular receptor interactions, and influencing cellular toxicity [50,157]. This multifunctionality complicates efforts to establish clear correlations between transfection efficiency and headgroup structure. However, for membrane fusion efficiency, the surface charge density of the lipid vector appears more critical than specific chemical features of the cationic lipid headgroup (Figure 4(ai)) [159,160]. While divalent cationic lipids outperform monovalent variants [36], excessive charge density can impair transfection [161,162], possibly by hindering nucleic acid release from the lipid carrier. Furthermore, high charge density can promote protein corona formation in biological environments, preventing direct contact between the lipid vector and cellular membrane (Figure 4(bi)) [163]. The theoretical model, incorporating hydration repulsion and electrostatic attraction between oppositely charged membranes [164], suggests that just 5 mol% of univalent charges suffices to maintain membrane proximity at approximately 1 nm, enabling spontaneous contact (Figure 4(ai)) [108,109]. This estimation aligns with experimental observations showing that lipid vectors containing low concentrations of positively charged lipids exhibit high fusogenicity with cell membranes and efficient transfection [89,165].

To ensure that electrostatic attraction between the lipid vector and the cellular membrane does not interfere with cell and tissue specificity, specific targeting strategies should be employed. Although these strategies are beyond the scope of this review, interested readers are encouraged to explore them in detail in [166,167]. The use of ionizable lipids, which become protonated in the pH environment characteristic of early endosomes, helps separate the vector’s specificity and fusogenicity functions. Additionally, a novel method to achieve tissue specificity by adjusting the charge profile of lipid vector membranes has been proposed [68,69].

PEGylated lipids are widely incorporated into lipid vectors, serving multiple essential functions: preventing vector aggregation [168], reducing toxicity [169], and extending blood circulation time [170]. However, like protein corona, the PEG coating creates steric hindrance that can impede membrane convergence and inhibit fusion processes (Figure 4(bii)). Studies have shown that even 2 mol% of PEGylated lipids in small unilamellar liposomes completely inhibits their fusion [171]. To address this limitation, recent formulations have reduced PEGylated lipid concentrations [158]. For example, Onpattro (patisiran), an FDA-approved lipid vector-based siRNA therapeutic for transthyretin-induced amyloidosis (hATTR), utilizes just 1.5 mol% PEGylated lipids [172]. An alternative strategy employs short-chain PEGylated lipids that gradually dissociate from the vector surface [173]. This approach maintains PEGylation’s initial benefits during circulation while allowing sufficient PEG clearance by the time vectors reach the endosome, enabling membrane fusion [168].

Beyond electrostatic interactions, “molecular zip” technologies can promote tight membrane contact between vectors and cells (Figure 4a) [174]. These approaches utilize complementary binding pairs strategically positioned on the fusion compartment surfaces, including DNA–DNA hybridization (Figure 4(aii)) [175] and coiled-coil peptide pairs (Figure 4(aiii)) [176]. For instance, the insertion of short (10–20 nucleotides) complementary DNA strands conjugated with lipid molecules into the membrane can drive specific membrane docking [177] and catalyze membrane fusion [178], while synthetic peptide pairs like E3/K3 demonstrate efficient membrane fusion in model systems [179]. To achieve effective membrane fusion, using proper coiled-coil formation between complementary peptides is essential. Analysis of 10 SNARE-like peptide pairs revealed that despite careful positioning of hydrophobic and electrostatic amino acids to promote interpeptide attraction, some motif pairs failed to achieve the necessary coiled-coil conformation, resulting in reduced fusogenic activity [180]. Critically, maintaining this optimal peptide conformation depends on proper membrane anchoring—cholesterol moieties have proven particularly effective, enabling spontaneous bilayer integration while preserving the peptides’ fusogenic structure [181]. When these design requirements are met, such systems demonstrate remarkable fusion specificity and control, capable of merging thousands of small liposomes into giant vesicles with fusion rates proportional to fusogen concentration [182]. While “molecular zip” technologies show excellent efficiency in controlled environments and hold promise for in vitro applications, their utility for in vivo delivery remains limited due to potential immunogenicity and stability concerns in biological fluids.

### 4.2. Optimizing Surface Dynamics and Hydrophobic Defects in Lipid Vectors

The initial step in successful membrane fusion requires establishing close proximity between the lipid vector shell and cell membrane, followed by the formation of hydrophobic contacts. This process is driven by hydrophobic attraction, where surfaces with low water affinity undergo rapid dehydration and aggregation. Success depends on both membrane participants: the dynamic nature of the vector’s lipid shell must allow the formation of transient hydrophobic defects, while the cellular membrane’s inherent heterogeneity (including lipid rafts and protein clusters) creates fusion-susceptible regions (Figure 5) [183,184]. Importantly, neither the vector’s lipid envelope nor the cell membrane exists in a steady state—both interfaces undergo continuous dynamic reorganization driven by cellular maintenance processes and environmental factors such as pH, temperature, ionic strength, and membrane-active molecules. This dynamic nature of both interfaces can either facilitate or hinder fusion events, depending on local conditions and timing [185].

Hydrophobic defects in lipid membranes can be promoted through several distinct mechanisms. The incorporation of cone-shaped lipids such as DOPE or cholesterol creates areas of membrane stress and packing defects due to their non-cylindrical molecular geometry (Figure 5ii) [135,186]. Complementarily, unsaturated lipids increase membrane fluidity, thereby enhancing both the frequency of hydrophobic defect formation and the capacity for local membrane reorganization [148,187]. The beneficial effect of unsaturated lipids on transfection efficiency has been consistently demonstrated across multiple studies [188,189], likely due to their ability to promote membrane–membrane interactions through enhanced flexibility and increased probability of hydrophobic defect formation.

The frequency and distribution of hydrophobic defects on lipid vector membrane surfaces can be precisely controlled through formulation design. One effective approach utilizes temperature-sensitive compositions exhibiting solid–liquid crystalline phase coexistence at physiological temperature (37 °C) (Figure 5i). This strategy has been extensively validated across various lipid systems, including ethylphosphatidylcholine mixtures with oleoyl, myristoyl and palmitoyl chains (EDOPC, EDMPC, EDPPC), quaternary ammonium lipids (diC14DAB, diC18DAB), dimyristoyltrimethylammonium-propane (DMTAP), and their combinations such as EDMPC/EDPPC, diC14DAB/diC18DAB, EDOPC/DMTAP, and EDOPC/diC14DAB [190]. Differential scanning calorimetry (DSC) and small-angle X-ray scattering (SAXS) analyses confirm phase transitions, while fluorescence resonance energy transfer (FRET) assays demonstrated superior membrane fusion activity in phase-coexistent formulations compared to single-phase systems.

Another sophisticated approach employs pH-responsive ionizable lipids that undergo conformational changes in acidic environments, particularly within endosomes (pH 5.5–6.5). A notable recent advancement features a lipid architecture combining a pH-switchable zwitterionic headgroup with three hydrophobic tails [191]. Under acidic conditions, this molecule adopts a cone shape, promoting membrane destabilization through both geometric and charge-based mechanisms. This pH-dependent conformational switching enables precise temporal and spatial control over the fusion process within the cellular trafficking pathway, resulting in significantly enhanced transfection efficiency compared to traditional cationic lipid formulations.

The importance of hydrophobic moiety design in lipid vectors is further emphasized by comparative studies of clinically approved ionizable cationic lipids. Analysis of four such lipids (Dlin-MC3-DMA, ALC-0315, SM-102, DODAP) revealed superior in vivo performance of ALC-0315 and SM-102 [192]. Notably, these superior performers share a distinctive structural feature—branched hydrophobic alkyl tails—suggesting that increased hydrophobic surface area available for potential water exposure may enhance fusion efficiency. This concept of optimizing hydrophobic surface interactions has been further explored through the incorporation of aromatic chromophore groups into lipid membranes. Remarkably, the addition of just ~5 mol% of lipids containing aromatic groups substantially increases fusogenic properties. Large polyaromatic hetero structures with significant induced dipole moments, particularly 4,4-difluoro-4-bora-3a,4a-diaza-s-indacene (BODIPY), indole, and rhodamine groups, demonstrated exceptional performance [90]. These findings were corroborated by studies using indole-containing cyanine-based gemini-like amphiphilic molecules (DiO and DiR), which showed enhanced liposomal fusogenicity when combined with PE and DOTAP across various cell types [193]. The fusogenic mechanism of these aromatic-containing lipids, while not fully understood, likely stems from their unique structural architecture. The polyaromatic π-conjugated fragments maintain significant lipophilic character despite their headgroup location and surface exposure. This creates an unconventional amphiphilic structure where lipophilic domains exist within the typically hydrophilic headgroup region. These exposed lipophilic areas presumably function as hydrophobic defects (Figure 5iii), creating energetically favorable interaction sites that lower the energy barrier for membrane fusion by reducing hydration repulsion between approaching membranes.

### 4.3. From Membrane Contact to Structural Reorganization

Following initial physical contact between the proximal monolayers of the cell membrane and lipid vector, local molecular rearrangements must occur to initiate fusion. The connection between hydrophobic defects in these opposing monolayers should spontaneously transform into intermediate fusion structures requiring minimal activation energy—first into a stalk (Figure 5). This transformation necessitates localized destabilization of the bilayer structure. The propensity of lipids to undergo such rearrangements depends on their ability to accommodate the geometry of intermediate structures and their initial packing stress in the vector’s outer monolayer (see Section 3).

Conical lipids with negative spontaneous curvature (DOPE, cholesterol) facilitate this process by increasing H_II_ phase transition propensity and creating beneficial packing stress (Figure 6i) [46,155]. This importance is exemplified in DOTAP/DOPE complexes with nucleic acids, wherein the transfection efficiency remains independent of DOTAP concentration [162] and is thus related to the readily occurring fusion of the membranes of DOTAP/DOPE lipid vectors with the cell’s plasma and endosomal membranes, as confirmed by 3D confocal microscopy [161]. Consequently, DOPE and cholesterol have become essential components in most commercial lipid-based transfection products, including Lipofectamine, MegaFectin, Lipofectin [155,194,195,196]. Similarly, the incorporation of ionizable lipids with branched or multiple alkyl chains, where the cross-section of the hydrophobic portion exceeds that of the hydrophilic head, enhances lipofection efficiency by promoting both the formation of inverted micellar aggregates (facilitating DNA encapsulation) and membrane fusion [50,197,198]. Moreover, membrane packing stressing may be tied to pH-responsivity by designing specific lipids that undergo conformational changes upon protonation, favoring more negative curvature with the protonated form [191,199].

The conical shape critical for membrane fusion can be acquired when ionizable lipids mix with negatively charged lipids of the cell membrane. The formation of ion pairs between oppositely charged lipid heads leads to their condensation, significantly reducing the total effective area of the polar region. This reduction in polar head area relative to the hydrophobic portion effectively creates negative spontaneous curvature [87]. This mechanism is believed to drive the destabilization of the cell membrane’s lamellar structure by cationic or ionizable lipids [200].

Efficient mixing between the proximal monolayers of the cell and vector membranes following hydrophobic contact requires a substantial chemical potential gradient between these membranes (Figure 6ii). This gradient can be established through several key differences: (a) the vector membrane can be engineered with distinctly different lipid compositions compared to cellular membranes, creating a thermodynamic drive for lipid mixing upon contact (compositional asymmetry); (b) strategic incorporation of cationic or ionizable lipids in the vector membrane creates electrostatic potential differences relative to the predominantly negative charge of cellular membranes, promoting rapid lipid redistribution upon contact (charge distribution); (c) vector membranes can be designed with inherent packing stress through the incorporation of non-bilayer lipids or curved geometries, creating a higher energy state that seeks relief through mixing with the less stressed cellular membrane structure (elastic strain [136]). The magnitude of these chemical potential differences directly determines the rate of lipid flux between membranes—higher differences generate stronger thermodynamic driving forces, leading to more rapid lipid mixing and subsequent membrane reorganization once initial hydrophobic contact is established (Figure 6, formation of hemifusion diaphragm), thereby promoting the fusion process. Moreover, membrane fluidity plays a crucial role in supporting effective lipid mixing and membrane structural reorganization from local contact to the hemifusion diaphragm again making the incorporation of unsaturated lipids particularly beneficial for lipofection [188,189,197].

### 4.4. Lipoplex Disassembly and Nucleic Acid Liberation

The final stage of gene delivery via lipid vectors requires the release of DNA following complete membrane fusion. This critical step involves several sequential processes that begin with the formation and expansion of fusion pores in the hemifusion diaphragm (Figure 3a). These pores emerge through the merging of distal monolayers (trans-leaflets) of the vector’s lipid envelope and cellular membrane, creating direct continuity between the vector’s interior and cellular cytoplasm. When discussing the enhancement of transfection efficiency through increased vector fusogenicity, attention typically focuses on three initial stages—attachment, contact formation, and local structural transition from lamellar to hexagonal phase. However, the final stage of DNA liberation often receives insufficient consideration, despite potentially representing a thermodynamic bottleneck in efficient genetic material delivery to the cytoplasm. The formation of fusion pores is energetically costly and could represent a key rate-limiting step in DNA liberation, further hampered by strong electrostatic associations between cationic lipids and negatively charged nucleic acids. Moreover, while the incorporation of inverted conical lipids (small heads, large tails) promotes initial fusion by destabilizing the proximal monolayer, it may paradoxically complicate fusion pore formation by stabilizing the distal DNA-facing monolayer, creating opposing effects at different stages of the delivery process.

The thickness of the central bilayer of the hemifusion diaphragm emerges as a critical determinant of its susceptibility to pore formation and subsequent DNA or RNA release. Molecular dynamics simulations reveal that the energy barrier for pore nucleation increases significantly with membrane thickness [98,128]. The reduced thickness and altered molecular packing of the hemifusion diaphragm play crucial roles in fusion pore nucleation. As the diaphragm thins, the energy barrier for pore formation decreases, primarily due to the shorter distance that must be bridged to create a continuous aqueous connection [95]. Local fluctuations in this thinner, less structured membrane more readily nucleate fusion pores, ultimately leading to complete membrane fusion (Figure 7i).

This intrinsic relationship between diaphragm thickness and fusion pore formation suggests that strategies targeting diaphragm properties represent a promising approach for enhancing lipofection efficiency. The composition of the hemifusion diaphragm is particularly critical, as one of its monolayers derives from lipids surrounding the polynucleic acid cargo and is thus enriched with cationic or ionizable lipids (Figure 6). Consequently, the acyl chain length of these lipids plays a crucial role in transfection efficiency.

Extensive experimental studies consistently demonstrate the superior performance of lipid vectors incorporating cationic or ionizable lipids with shorter alkyl chains [201,202]. For example, systematic investigations of ethyl-phosphatidylcholine (EPC) derivatives reveal that reducing acyl chain length from 18 to 14 carbons results in an approximately 10-fold increase in transfection efficiency [203]. This pattern holds across different lipid classes: hydroxyethyl quaternary ammonium lipids with di-C14 chains demonstrate superior performance compared to their di-C16 and di-C18 counterparts [204]; synthetic cationic glutamide lipids show optimal performance with 12-carbon alkyl tails compared to 14- and 16-carbon variants [205].

However, optimization requires careful balance, as extremely short acyl chains may compromise vector stability and cellular uptake. Thus, effective vector design must reconcile two competing requirements: shorter chains for efficient fusion pore formation versus sufficient hydrophobic interactions for maintaining vector integrity during cellular delivery [80].

A promising strategy to enhance fusion pore formation in the hemifusion diaphragm involves incorporating lipids with positive spontaneous curvature (characterized by a large headgroup-to-tail ratio. These molecules preferentially accumulate in regions of high positive membrane curvature, particularly at the rim of nascent fusion pores, thereby reducing the energy barrier for pore formation and expansion (Figure 7ii) [206].

Ionizable lipids can effectively serve this curvature-inducing role through pH-dependent mechanisms. The acidic environment of endosomes (pH 5.5–6.5) provides an opportunity to trigger positive curvature formation at the appropriate cellular location and time. Upon exposure to this acidic environment, ionizable lipids with suitable pKa values undergo protonation [55,63,87], which increases their effective headgroup size through enhanced solvation and electrical repulsion between neighboring charged groups [207].

These ionizable lipids create beneficial asymmetry between the lipid vector’s leaflets required for efficient fusion with cellular membranes through distinct mechanisms in each monolayer [145,146]. In the proximal (outer) monolayer, the formation of ion pairs with cellular anionic lipids creates molecular complexes with inverted conical shapes, facilitating local H_II_ phase transition. In the distal monolayer (facing polynucleic acid), increased headgroup size promotes positive curvature necessary for pore formation in the hemifusion diaphragm.

However, a significant limitation exists: the strong electrostatic interactions between these lipids and the negatively charged polynucleic acid cargo can restrict the conformational changes necessary for fusion promotion. This highlights the need for careful optimization of lipid pKa values and vector composition to balance nucleic acid binding with fusion-promoting capabilities [87].

One effective strategy to overcome this limitation involves introducing polymers that compete with lipids for DNA binding, such as polyethyleneimine (PEI). In this approach, PEI forms a condensed core complex with DNA, while lipids create an outer envelope structure. This architecture maintains the fusogenic properties of the lipid envelope while effectively condensing the genetic cargo. Multiple experimental studies have confirmed the synergistic enhancement of gene delivery efficiency when combining PEI and cationic lipids [208,209,210]. In these hybrid systems, PEI is suggested to provide strong DNA condensation while the lipid envelope protects genetic cargo and facilitates membrane fusion and cellular entry. The effectiveness of this approach is demonstrated by the combination of PEI (25 kDa) with the DOTAP–cholesterol mixture, which leads to a remarkable 10–100-fold increase in transfection efficiency compared to pure lipid systems [209].

Similar enhancements can be achieved using alternative polycationic molecules. Chitosan, like PEI, effectively condenses DNA and RNA, and its addition to lipid formulations significantly increases their transfection efficiency [211,212]. This versatility in polymer choice provides opportunities for optimizing vector properties based on specific application requirements.

The conjugation of cationic lipids with membrane-penetrating peptides represents an innovative strategy to enhance lipoplex fusion and cargo release (Figure 7ii). These hybrid molecules combine the DNA condensation capabilities of cationic lipids with the membrane-destabilizing properties of specific peptide sequences. The HIV-derived TAT peptide (YGRKKRRQRRR), when conjugated to PEGylated-PE lipids, significantly enhances cellular uptake and endosomal escape of liposomes containing small amounts of cationic lipid DOTAP (<10 mol%). These TAT-modified formulations demonstrate high transfection efficiency while maintaining remarkably low cytotoxicity [213,214]. Similar improvements in gene delivery have been achieved using arginine-rich dendritic cationic lipopeptides, which combine the advantages of branched architecture with membrane-penetrating capabilities [80,215] The effectiveness of arginine-based modifications is further supported by studies showing that polyarginine fragments themselves can act as potent cell-penetrating peptides, facilitating both cellular entry and endosomal escape [216].

The mechanism of enhanced fusion likely involves direct perturbation of the hemifusion diaphragm structure. When present in the distal monolayer of the lipid vector, these membrane-penetrating peptides can be inserted into the hemifusion diaphragm, creating local packing defects in its highly ordered lipid structure. These defects can act as nucleation sites for pore formation, lowering the energy barrier for pore opening and expansion. The amphipathic nature of these peptides, combining hydrophobic residues that insert into the lipid phase with hydrophilic segments that prefer an aqueous environment, makes them particularly effective at destabilizing the planar bilayer structure of the hemifusion diaphragm (Figure 7ii). This mechanism provides an additional pathway for fusion pore formation, complementing other fusion-promoting factors such as membrane tension and lipid spontaneous curvature.

## 5. Concluding Remarks and Future Perspectives

Membrane fusion represents a critical bottleneck in lipid-mediated gene delivery, with its efficiency directly impacting transfection success [84,89,91,193,217]. This review highlights how strategic manipulation of fusion mechanisms can dramatically enhance gene delivery outcomes. Several key principles emerge as particularly significant for vector design.

Although most lipid-based vectors adopt non-lamellar structures, particularly when complexed with nucleic acids [44,45,46,47,48], the fundamental principles of bilayer fusion remain highly relevant to understanding and optimizing their fusion mechanisms. The initial stages of fusion, from membrane contact through stalk formation, follow remarkably similar pathways in both lamellar and non-lamellar systems, as demonstrated by molecular dynamics simulations [98]. While the subsequent stages might differ in their precise geometric arrangements, the basic requirement for a merger of proximal lipid monolayers persists regardless of the initial structure (Figure 4 and Figure 5). In non-lamellar systems, the expansion of initial stalks necessarily leads to structures analogous to hemifusion diaphragms, as the distal monolayers must ultimately come into contact to complete the fusion process. This mechanistic similarity allows us to apply the wealth of knowledge gained from studying classical membrane fusion to the design and optimization of non-lamellar lipid vectors.

The multistep nature of membrane fusion demands precise orchestration of molecular events, from initial membrane contact requiring overcome of electrostatic and hydration barriers, through stalk formation depending on local membrane destabilization and lipid mixing, to hemifusion diaphragm formation and expansion involving complex lipid reorganization, and finally to fusion pore nucleation and expansion requiring specific molecular architectures. Each stage presents unique challenges but also opportunities for optimization through rational vector design.

Vector architecture must balance multiple, often competing, requirements. Surface properties must promote specific cellular targeting while enabling membrane contact, achieved through careful control of charge density and PEGylation. Lipid compositions need to facilitate both initial fusion through negative curvature lipids (Figure 5 and Figure 6) and subsequent pore formation through positive curvature components (Figure 7). Structural stability during circulation must be maintained while ensuring efficient cargo release at the target site, and DNA condensation should be strong enough for protection but allow timely release. These competing demands necessitate sophisticated design strategies, often involving stimuli-responsive components.

Importantly, these opposing requirements typically apply to different monolayers of the vector: negative spontaneous curvature is needed to facilitate the fusion of its outer monolayer, while positive spontaneous curvature is beneficial for the rupture of the hemifusion diaphragm composed of the distal monolayers of fusing membranes. This spatial separation of functions allows partial resolution of these competing demands through the assembly of compositionally asymmetric lipid monolayers, although such asymmetry significantly complicates production technology.

Several promising strategies have emerged for enhancing fusion efficiency, with structural modifications of lipids playing a critical role in determining the mechanisms and efficiency of lipofection. These include precise control of surface charge density [89,161,162,165] and dynamic PEGylation [29,168,173], where changes to the lipid headgroup, such as introducing zwitterionic or PEGylated moieties, can modulate cellular uptake and endosomal escape. Strategic incorporation of fusogenic lipids with negative spontaneous curvature (DOPE, cholesterol) [155,162,186,193,195,196,209] and variations in fatty acid chain length and saturation can influence membrane fluidity, fusogenicity, and complex stability. Particularly noteworthy are advances in ionizable lipid design [87,218,219], where modifications to the headgroup structure and pKa values enable pH-dependent charge switching, promoting efficient DNA complexation at low pH while minimizing cytotoxicity at physiological pH [201,220]. For instance, the incorporation of tertiary amines with optimized pKa (~6.5) or pH-sensitive linkers between the headgroup and hydrophobic domains has led to significant improvements in transfection efficiency [221]. Similarly, modifications to cationic lipid structure, such as incorporating biodegradable ester linkages or optimizing the spacing between charged centers, have enhanced gene delivery while reducing toxicity [55]. The use of pH-responsive components for environmental triggering in endosomes [191], integration of membrane-penetrating peptides for enhanced fusion and cargo release [213,214,215], and polymer–lipid hybrid systems combining efficient DNA condensation with fusion capabilities [208,209,210,211,212] further demonstrate how lipid engineering can optimize transfection outcomes. However, these modifications must be balanced against potential increases in cytotoxicity or reduced biodistribution stability. Each approach offers distinct advantages and can be combined for synergistic effects, highlighting the importance of rational lipid design for optimizing transfection efficiency.

The final stages of fusion, particularly pore formation and cargo release, deserve greater attention in vector design. Hemifusion diaphragm thickness emerges as a critical parameter for pore formation, while acyl chain length of cationic lipids significantly impacts fusion pore formation. DNA–lipid interactions can either promote or inhibit final cargo release, and strategic placement of positive curvature components in distal monolayers enhances pore formation.

While not discussed in detail in this review, cell-derived ghost vectors represent a promising new direction in fusion-dependent gene delivery. These vectors, derived from natural cell membranes through hypotonic lysis and careful reconstitution, maintain many of the fusion-promoting proteins and lipid organizations found in cellular membranes [72,222,223]. Ghost technology offers several unique advantages: natural membrane composition optimized for fusion, presence of SNARE proteins and other fusion machinery, reduced immunogenicity compared to synthetic vectors, and potential for tissue-specific targeting through source cell selection [73,224]. Recent advances in ghost preparation and modification techniques suggest their potential as a next-generation platform for gene delivery, particularly for applications requiring precise tissue targeting or reduced immune response [78,79].

Looking forward, several areas warrant further investigation. These include development of “smart” vectors with environmentally triggered fusion capabilities, including new pH-sensitive lipids and conformational switches; better understanding of the relationship between DNA topology and fusion dynamics, particularly in the context of different genetic cargo types; design of components that can differentially influence proximal versus distal monolayer behavior for optimized fusion progression; integration of multiple fusion-enhancing strategies without compromising vector stability; advancement of ghost technology through improved preparation methods and targeted modifications; and development of quantitative methods for measuring fusion efficiency in living cells.

These insights provide a framework for the rational design of next-generation lipid vectors with enhanced fusion capabilities and, consequently, improved gene delivery efficiency. The field continues to evolve rapidly, with a new understanding of fusion mechanisms enabling increasingly sophisticated vector designs.

## Figures and Tables

**Figure 1 ijms-25-13540-f001:**
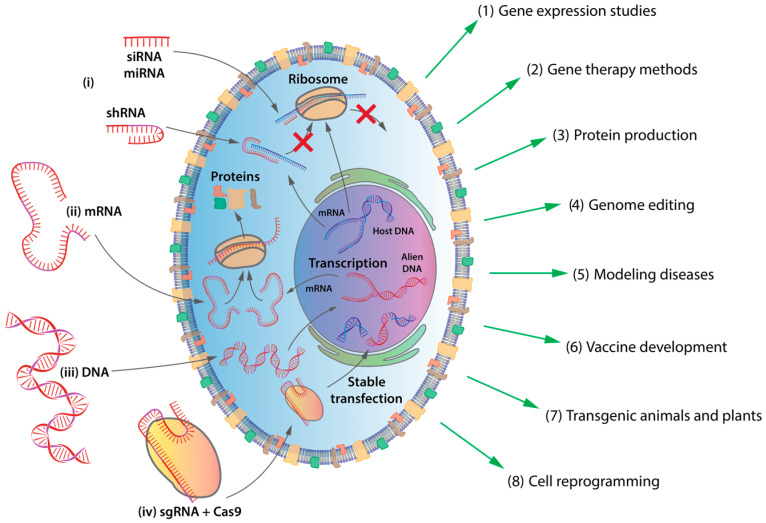
Applications of transfection in biological systems. (**i**) Gene silencing via RNA: transfected siRNA, miRNA, or antisense RNA interfere with mRNA translation, leading to gene silencing (e.g., studying gene function, therapeutic target validation). (**ii**) mRNA delivery and translation: transfected mRNA is translated into proteins by ribosomes (e.g., protein expression studies, therapeutic protein production). (**iii**) Gene addition and expression: recombinant transgenic, “alien” DNA introduced into the nucleus undergoes transcription, resulting in mRNA production, which subsequently enters the cytoplasm for translation (e.g., overexpression studies, generating cell lines with specific characteristics). (**iv**) Stable transfection: alien DNA integrates into the host genome (often mediated by CRISPR/Cas9 systems), enabling long-term expression (e.g., generation of stable cell lines for research and therapeutic applications).

**Figure 2 ijms-25-13540-f002:**
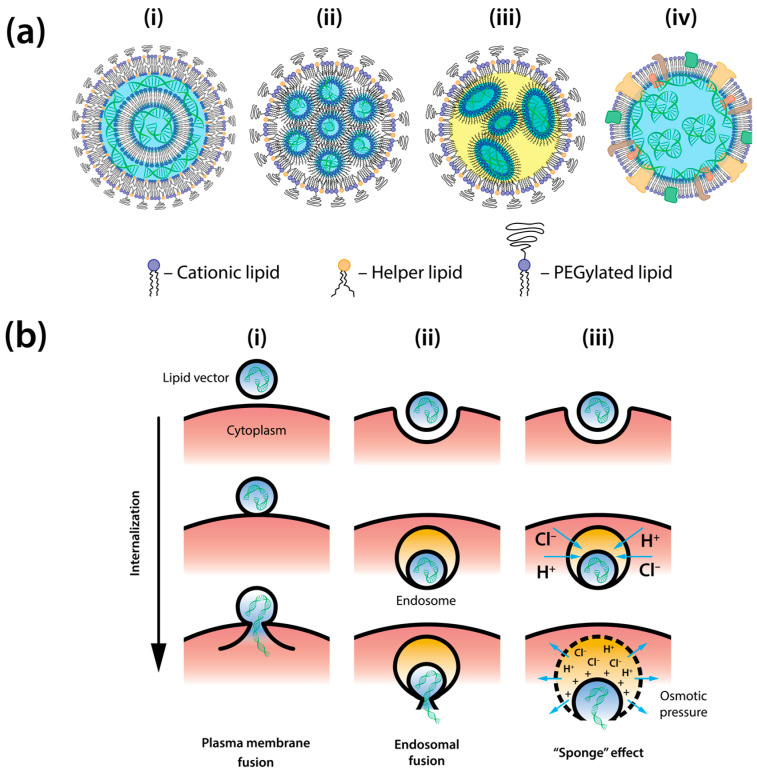
Structure and cellular entry mechanisms of lipid-based gene delivery vectors. (**a**) Structural diversity of lipid-based vectors for gene delivery: (**i**) Lamellar lipoplex: multilamellar structure with DNA/RNA sandwiched between cationic lipid bilayers. (**ii**) Hexagonal phase lipoplex: inverted hexagonal phase with nucleic acids enclosed within lipid-lined water channels. (**iii**) Solid lipid nanoparticle (SLN)/nanostructured lipid carrier (NLC): solid or solid–liquid matrix core surrounded by lipid monolayer. (**iv**) Nanoghost: cell membrane-derived vesicle retaining native membrane proteins and incorporating genetic cargo. (**b**) Pathways for cellular entry and genetic cargo delivery. (**i**) Direct fusion pathway: immediate fusion with plasma membrane; direct cytoplasmic release of genetic material; bypasses endosomal compartmentalization. (**ii**) Endosomal fusion escape: internalization via endocytosis; fusion with endosomal membrane; controlled release of genetic cargo; membrane merger preserves compartment integrity. (**iii**) Endosomal rupture pathway: pH-dependent ionization of lipids; osmotic pressure buildup (“proton sponge effect”); endosomal membrane destabilization and rupture; bulk release of vesicle contents.

**Figure 3 ijms-25-13540-f003:**
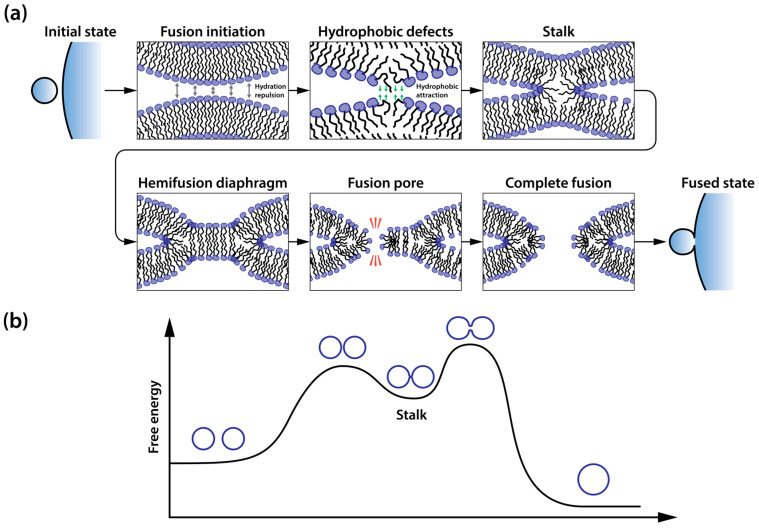
Membrane fusion pathway and associated energy landscape. (**a**) Sequential stages of membrane fusion. Key feature shown in cross-sectional view. (**b**) Free energy profile of the fusion cascade.

**Figure 4 ijms-25-13540-f004:**
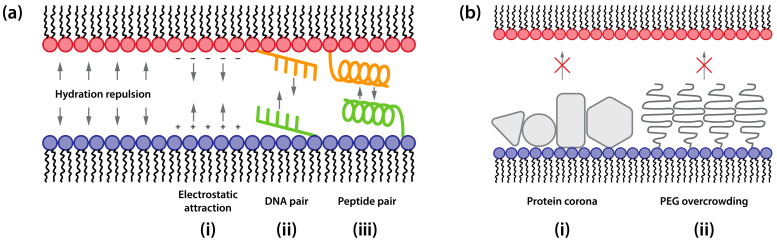
Factors modulating lipid-based vector approaching cellular membrane. (**a**) Strategies enhancing vector-membrane adhesion. (**b**) Barriers impeding vector-membrane contact.

**Figure 5 ijms-25-13540-f005:**
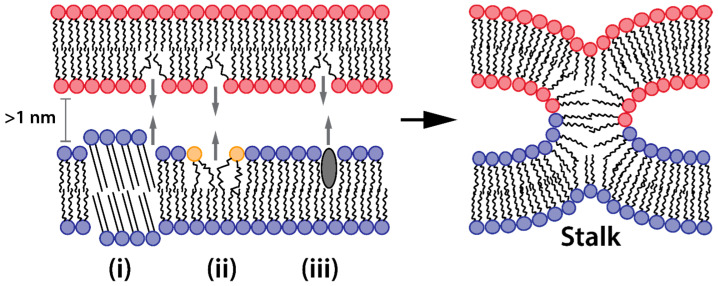
Sources of hydrophobic defect formation in lipid-based vector shell: (**i**) solid–liquid crystalline phase coexistence; (**ii**) lipid packing defects produced by inverted conical lipids; (**iii**) lipophilic moieties in lipid-based vector formulation.

**Figure 6 ijms-25-13540-f006:**
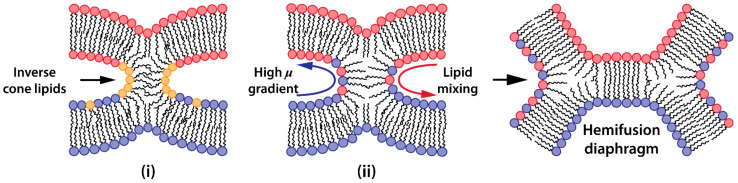
Driving forces of membrane reorganization after its initial contact: (**i**) the accumulation of inverted-conical lipids (indicated in yellow) that facilitate highly curved temporary structures; (**ii**) lipid mixing under high chemical potential gradient blue and red arrows indicate the flux direction of lipids from different membranes, shown as red and blue circles.

**Figure 7 ijms-25-13540-f007:**
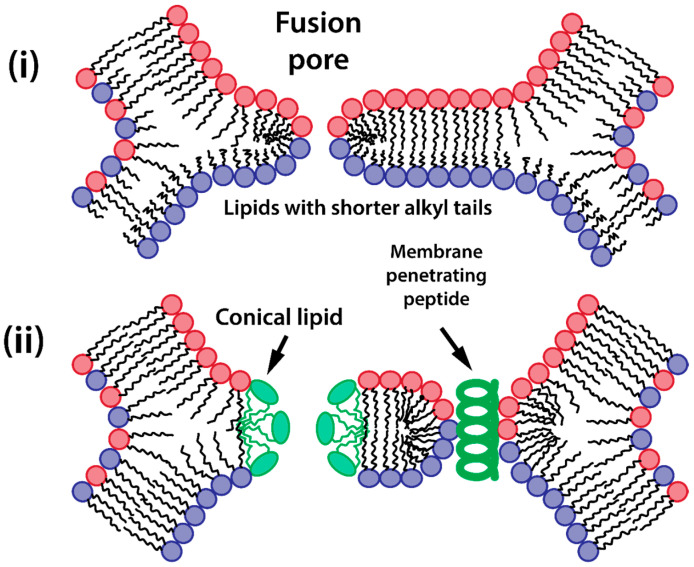
Strategies for enhanced pore formation in hemifusion diaphragm: (**i**) diaphragm thinning by employment of cationic lipids with shorter acyl chains; (**ii**) ionizable lipids having cone shape in protonated state. Red and blue circles indicate lipids from two different membranes.

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
