# Peer review of "Boosting Lipofection Efficiency Through Enhanced Membrane Fusion Mechanisms"

_ijms, 2024, doi:10.3390/ijms252413540_

Round 1
Reviewer 1 Report
Comments and Suggestions for Authors
Bashkirov and coworkers systemically reviewed the mechanism of using lipid vectors for transfection through membrane fusion. This article is nicely written with a detailed discussion of fundamental understandings of the system, which inspires the readers while reading. The only comment is that adding some discussion on how structural modification of lipids can influence the mechanisms and final transfection outcomes will be beneficial to the audience.
Author Response
Comment:
Bashkirov and coworkers systemically reviewed the mechanism of using lipid vectors for transfection through membrane fusion. This article is nicely written with a detailed discussion of fundamental understandings of the system, which inspires the readers while reading. The only comment is that adding some discussion on how structural modification of lipids can influence the mechanisms and final transfection outcomes will be beneficial to the audience.
Authors' reply:
We are very grateful for kind words and suggestions regarding our paper.
We totally agree with the reviewer. Additional discussion about structural modifications of lipids would be an improvement to the text indeed. We added the following discussion including new references in section 5. Concluding remarks and future perspectives, and marked it with blue type:
Several promising strategies have emerged for enhancing fusion efficiency, with structural modifications of lipids playing a critical role in determining the mechanisms and efficiency of lipofection. These include precise control of surface charge density [89,161,162,165] and dynamic PEGylation [29,168,173], where changes to the lipid headgroup, such as introducing zwitterionic or PEGylated moieties, can modulate cellular uptake and endosomal escape. Strategic incorporation of fusogenic lipids with negative spontaneous curvature (DOPE, cholesterol) [155,162,186,193,195,196,209] and variations in fatty acid chain length and saturation can influence membrane fluidity, fusogenicity, and complex stability. Particularly noteworthy are advances in ionizable lipid design [87,218,219], where modifications to the headgroup structure and pKa values enable pH-dependent charge switching, promoting efficient DNA complexation at low pH while minimizing cytotoxicity at physiological pH [201,220]. For instance, incorporation of tertiary amines with optimized pKa (~6.5) or pH-sensitive linkers between the headgroup and hydrophobic domains has led to significant improvements in transfection efficiency [221]. Similarly, modifications to cationic lipid structure, such as incorporating biodegradable ester linkages or optimizing the spacing between charged centers, have enhanced gene delivery while reducing toxicity [55]. The use of pH-responsive components for environmental triggering in endosomes [191], integration of membrane-penetrating peptides for enhanced fusion and cargo release [213–215], and polymer-lipid hybrid systems combining efficient DNA condensation with fusion capabilities [208–212] further demonstrate how lipid engineering can optimize transfection outcomes. However, these modifications must be balanced against potential increases in cytotoxicity or reduced biodistribution stability. Each approach offers distinct advantages and can be combined for synergistic effects, highlighting the importance of rational lipid design for optimizing transfection efficiency.
Reviewer 2 Report
Comments and Suggestions for Authors
Dear authors, Dear Editor
In the manuscript entitled “Boosting Lipofection Efficiency Through Enhanced Membrane Fusion Mechanisms” Bashkirov and co-workers review the literature regarding “the role of membrane fusion in determining the efficiency of lipid-based vectors and its potential to enhance gene transfection outcomes”.
I enjoyed reading this review. The review is well organized and clearly written and supported by a significant number of relevant references.
In the context of the recent development of lipidic vesicles as vehicles for delivery of RNAm to cells for vaccine applications, this review is timely and potentially appealing to a wide audience.
In its current state the review is a relevant update for specialists and a good entry point for young graduate researchers.
In my opinion, giving the review a “pedagogical” spin, could raise significantly its profile, appealing to a wider audience of “generalist” readers, from outside this research area. The authors should consider introducing more schemes, Figures and Tables, e.g. chemical structures of exquisite functional lipids and specialized membrane structures and mechanisms.
Best regards
Author Response
Comment:
In the manuscript entitled “Boosting Lipofection Efficiency Through Enhanced Membrane Fusion Mechanisms” Bashkirov and co-workers review the literature regarding “the role of membrane fusion in determining the efficiency of lipid-based vectors and its potential to enhance gene transfection outcomes”.
I enjoyed reading this review. The review is well organized and clearly written and supported by a significant number of relevant references.
In the context of the recent development of lipidic vesicles as vehicles for delivery of RNAm to cells for vaccine applications, this review is timely and potentially appealing to a wide audience.
In its current state the review is a relevant update for specialists and a good entry point for young graduate researchers.
In my opinion, giving the review a “pedagogical” spin, could raise significantly its profile, appealing to a wider audience of “generalist” readers, from outside this research area. The authors should consider introducing more schemes, Figures and Tables, e.g. chemical structures of exquisite functional lipids and specialized membrane structures and mechanisms.
Authors' reply:
We are very grateful for your interest in our work and kind words, and your suggestions.
We agree with the reviewer's point regarding the improvement of pedagogical aspect of our review. To make the review more reader-friendly, we decided to add a new figure with a scheme that describes the transfection process in general, and illustrates the application of transfection, in the Introduction. The figure shows different types of genetic material that can be input into the cell in order to achieve transient or stable transfection. It also rounds up multiple areas of research that strongly depend on transfection, such as gene editing, genetic research, protein production, etc. The figure is shown below, figure legend is marked by green type:

Figure 1. Applications of transfection in biological systems: (i) Gene silencing via RNA: Transfected siRNA, miRNA, or antisense RNA interfere with mRNA translation, leading to gene silencing (e.g., studying gene function, therapeutic target validation); (ii) mRNA delivery and translation: Transfected mRNA is translated into proteins by ribosomes (e.g., protein expression studies, therapeutic protein production); (iii) Gene addition and expression: Recombinant transgenic, “alien” DNA introduced into the nucleus undergoes transcription, resulting in mRNA production, which subsequently enters the cytoplasm for translation (e.g., overexpression studies, generating cell lines with specific characteristics); (iv) Stable transfection: Alien DNA integrates into the host genome (often mediated by CRISPR/Cas9 systems), enabling long-term expression (e.g., generation of stable cell lines for research and therapeutic applications).